# IPSI: Enhancing Structural Inference with Automatically Learned Structural Priors

**Zhongben Gong**    **Xiaoqun Wu**[*]    **Mingyang Zhou**
College of Computer Science and Software Engineering, Shenzhen University
zbgong1729@gmail.com, xqwu@szu.edu.cn, zmy@szu.edu.cn

## Abstract

We propose IPSI, a general iterative framework for structural inference in interacting dynamical systems. It integrates a pretrained structural estimator and a joint inference module based on the Variational Autoencoder (VAE); these components are alternately updated to progressively refine the inferred structures. Initially, the structural estimator is trained on labels from either a meta-dataset or a baseline model to extract features and generate structural priors, which provide multi-level guidance for training the joint inference module. In subsequent iterations, pseudo-labels from the joint module replace the initial labels. IPSI is compatible with various VAE-based models. Experiments on synthetic datasets of physical systems demonstrate that IPSI significantly enhances the performance of structural inference models such as Neural Relational Inference (NRI). Ablation studies reveal that feature and structural prior inputs to the joint module offer complementary improvements from representational and generative perspectives.

## 1 Introduction

In many domains, dynamical systems can be understood as collections of interacting agents, ranging from physical, biological to multi-agent systems [9, 5, 15, 13, 1, 10]. These interactions are often modeled by an interaction graph, where nodes represent agents and edges denote the existence of interactions. Understanding the structure of such a graph is essential for analyzing, controlling, and optimizing the behavior of the underlying system. However, in practice, the interaction structure is frequently unobserved or only partially known, and only the observable agent states are available. For instance, in molecular biology, understanding the interactions between drug compounds and target proteins is critical for applications such as drug discovery, side-effect prediction, and drug repurposing [6]. These interactions are typically governed by underlying molecular structures and biochemical affinities, which are difficult to deduce purely from theoretical analysis. Moreover, experimentally identifying all potential interactions is often costly and time-consuming. Structural inference offers a promising alternative by uncovering latent interaction patterns based on molecular dynamics that can be observed more easily, thereby reducing the reliance on expensive experimental procedures and providing mechanistic insights into pharmacological activity [15].

As a milestone in the field of structure inference, the Neural Relational Inference (NRI) model leverages the latent space of a Variational Autoencoder (VAE) [7] to model underlying structures [8]. However, it may face challenges when applied to complex physical systems such as charged particle interactions. This is largely due to the limitations of unsupervised joint training and the reliance on overly simplistic priors for latent variables. While some approaches incorporate prior structural knowledge, few systematically address how to conveniently obtain reliable structure priors across diverse dynamical systems.

---

[*]Corresponding author.

39th Conference on Neural Information Processing Systems (NeurIPS 2025).

To address these challenges, we propose the Iterative Pretrained Structural Inference Framework (IPSI), an iterative framework for structural inference that combines a pretrained structural estimator $\text{SI}_{\text{prior}}$ with a VAE-based joint inference module $\text{SI}_{\text{joint}}$. Specifically, $\text{SI}_{\text{joint}}$ receives embedding representations including structural information and a learnable structural prior both provided by $\text{SI}_{\text{prior}}$. These components, together with state labels, provide multi-level supervision for the various modules within $\text{SI}_{\text{joint}}$ during training. Meanwhile, $\text{SI}_{\text{joint}}$ and $\text{SI}_{\text{prior}}$ are alternately updated in an iterative process, allowing the model to escape local optima and progressively refine its structure inference.

$\text{SI}_{\text{prior}}$ can be seen as a structural prior network, and is trained under supervision using structural labels. To this end, we propose two complementary strategies to generate structural labels for the first iteration, while in subsequent iterations, pseudo-labels from $\text{SI}_{\text{joint}}$ are used instead. The design of IPSI follows a general paradigm common to many structure inference models, enabling seamless integration with a variety of existing frameworks. Experimental results demonstrate that IPSI significantly improves the performance of multiple baselines and achieves state-of-the-art accuracy across several datasets.

## 2 Related work

Structural inference aims to uncover latent interaction structures from observable sequences of agent states. Compared to traditional statistical and information-theoretic methods, recent deep learning approaches—particularly those based on neural networks—offer enhanced capabilities in modeling high-dimensional data requiring fewer assumptions. A major milestone in this direction is NRI, which pioneered the use of VAEs for inferring latent structures [8]. NRI employs a fully connected Graph Neural Network (GNN) encoder to propagate information across nodes, while modeling latent variables as probabilistic adjacency matrices. These latent structures are then used by the decoder to predict future states. However, NRI relies on oversimplified priors over the latent space and faces challenges in jointly training the encoder and decoder without external supervision—particularly in complex systems.

Building upon this foundation, some subsequent works have explored incorporating prior knowledge of real-world interaction structures into the NRI framework. Li et al. [11] and Chen et al. [2] introduced structural priors derived from real-world networks—such as degree distributions, sparsity, and connectivity—as regularization terms to guide latent structure learning. Along with these structural priors, Wang et al. [16] proposed an iterative training scheme in which edge weights are refined during training to emphasize likely interactions and suppress noisy connections. While these methods demonstrate improved structure inference, they often depend on manually crafted priors that are difficult to generalize across domains.

In addition to incorporating structural priors, several variants and extensions of NRI have been proposed to enhance performance and flexibility, including alternative decoding mechanisms [21], multi-interaction systems [22], and improved optimization strategies [4], among others [17, 23, 24, 18, 19]. Despite these advances, limited attention has been given to the question of how to obtain informative structural priors efficiently. This gap becomes critical in unsupervised settings, where the joint learning of dynamics and structure can suffer without effective prior guidance.

## 3 Preliminaries

### 3.1 Notations and Problem Formulation

In this paper, we denote the number of agents in the interacting system as $N$. The state of agent $i$ at time $t$ is represented by a vector $\mathbf{x}_i^t \in \mathbb{R}^d$, where $d$ is the dimensionality of the state space and the number of time steps is denoted as $T$. For simplicity, the collection of all agent states at time $t$ is denoted as $\mathbf{x}^t = \{\mathbf{x}_i^t\}_{i=1}^N \in \mathbb{R}^{N \times d}$, the time series data of agent $i$ is denoted as $\mathbf{x}_i = \{\mathbf{x}_i^t\}_{t=1}^T \in \mathbb{R}^{T \times d}$, and the collection of all time series data of all agents is denoted as $\mathbf{x} = \{\mathbf{x}_i\}_{i=1}^N \in \mathbb{R}^{N \times T \times d}$. The underlying interaction structure among the agents is modeled as a graph $\mathcal{G} = (\mathcal{V}, \mathcal{E})$, where $\mathcal{V} = \{v_1, v_2, \ldots, v_N\}$ represents the set of vertices (agents), and $\mathcal{E} \subseteq \mathcal{V} \times \mathcal{V}$ represents the set of edges encoding the interactions between agents. The interaction graph is characterized by its adjacency tensor $\mathbf{z} \in \mathbb{R}^{N \times N \times K}$ to consider multiple types of interactions,

where each entry $z_{ijk}$ is defined as:

$$z_{ijk} = \begin{cases} 1, & \text{if there is an interaction of type } k \text{ from agent } i \text{ to agent } j, \\ 0, & \text{otherwise.} \end{cases} \tag{1}$$

The problem of structural inference we consider in this paper is to recover the hidden interaction graph $\mathcal{G}$ or equivalently its adjacency tensor $\mathbf{z}$ from the observed sequences of agent states $\mathbf{x}$. More formally, given the agent states over time, the objective is to learn a function $f : \mathbf{x} \rightarrow \mathbf{z}$, where $f(\cdot)$ captures the underlying structure from states. This problem involves both inferring the structure of the graph $\mathcal{G}$ and modeling the interactions mechanisms that govern the dynamics of the system.

### 3.2 Neural Relational Inference

#### 3.2.1 Model Overview

NRI is a foundational method for unsupervised structure learning in dynamical systems, which learns the latent interaction graph among agents from observed trajectories without ground-truth edge information [8].

Built on the VAE framework, NRI consists of two main components: a GNN-based encoder and a trajectory-prediction decoder. The encoder processes observable node sequences $\mathbf{x} \in \mathbb{R}^{N \times T \times d}$ through message passing to extract structural information and outputs a distribution over latent edge types, modeled as categorical variables with $K$ classes, forming the adjacency tensor $\mathbf{z} \in \mathbb{R}^{N \times N \times K}$:

$$\mathbf{h} = f_{\text{enc}}(\mathbf{x}), \quad q_\phi(\mathbf{z}|\mathbf{x}) = \text{softmax}(\mathbf{h}). \tag{2}$$

Since directly sampling the discrete adjacency tensor $\mathbf{z}$ is non-differentiable, NRI employs the Gumbel-Softmax trick [12] for backpropagation. The inferred graph $\mathbf{z}$ is then fed into the GNN-based decoder, which predicts future dynamics via message passing on the sampled interaction graph. The decoder is optimized to minimize reconstruction loss between predicted and true future states:

$$\mathbf{z}ij = \text{softmax}\left((\mathbf{h}ij + \mathbf{g})/\tau\right), \quad p_\theta(\mathbf{x}|\mathbf{z}) = \prod_{t=1}^{T} p_\theta(\mathbf{x}^{t+1}|\mathbf{x}^{1:t}, \mathbf{z}). \tag{3}$$

The model is trained by maximizing the evidence lower bound (ELBO):

$$\text{ELBO} = \mathbb{E}q\phi(\mathbf{z}|\mathbf{x})[\log p_\theta(\mathbf{x}|\mathbf{z})] - D_{\text{KL}}(q_\phi(\mathbf{z}|\mathbf{x})||p(\mathbf{z})), \tag{4}$$

where the first term minimizes state prediction error and the second regularizes the latent space by constraining $q_\phi(\mathbf{z}|\mathbf{x})$ to align with the prior $p(\mathbf{z})$ (typically uniform).

#### 3.2.2 Limitations of NRI

Despite the elegant formulation and general applicability of NRI, it suffers from several notable limitations when applied to complex dynamical systems. First, the uniform prior imposed over the latent interaction graph often deviates significantly from the true underlying structure which leads to suboptimal inference. Second, the joint unsupervised training of encoder and decoder relies solely on trajectory reconstruction loss, which may not provide sufficiently informative gradients to uncover accurate interaction structures. Lastly, the absence of external structural supervision makes one-shot training strategies prone to premature convergence to suboptimal solutions. These limitations highlight the need for a more flexible and informed structure inference framework—one that can incorporate external structural cues to provide multi-level supervision or guidance for both the encoder and decoder, and employ iterative optimization to escape local minima.

## 4 Model design

### 4.1 Motivation and Overall Pipeline Architecture

Most VAE-based structural inference models can be summarized as follows (trajectory embedding is separated from the encoder):

$$\text{trajectory data} \xrightarrow{\text{embedding}} \text{embedding vector} \xrightarrow{\text{encoder}} \text{inferred structure} \xrightarrow{\text{decoder}} \text{predicted trajectory}$$

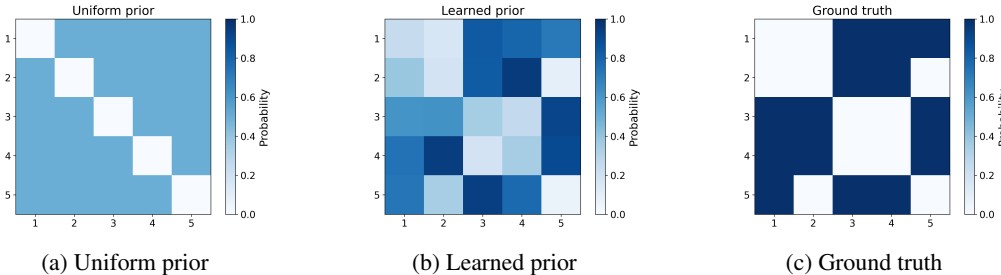

Figure 1: Comparison of two kinds of prior distributions and the ground truth.

This paradigm consists of three trainable components: trajectory embedding, encoder, and decoder. However, only the ground-truth trajectories are available as supervision signals. In IPSI, we address this issue by providing pretraining inputs for the trajectory embedding and encoder modules. These two inputs are used to guide the embedding and encoder modules, and together with state labels, achieving simultaneous guidance of three trainable modules, thus effectively alleviating the difficulties caused by long chain joint training. Specifically, the pretrained structure estimator $\text{SI}_{\text{prior}}$ extracts features from raw trajectories. These features are concatenated with the original node embeddings and fed into the encoder of $\text{SI}_{\text{joint}}$. Since the training of $\text{SI}_{\text{prior}}$ is supervised by structural labels, this strategy will result in richer representations that embed structural cues.

Next, $\text{SI}_{\text{prior}}$'s predicted edge probabilities define an informed prior $q_\phi^{\text{prior}}(\mathbf{z}|\mathbf{x})$ of structure. We replace the standard uniform prior $p(\mathbf{z})$ in the KL divergence term with this learned prior:

$$\mathcal{L}_{\text{KL}} = D_{\text{KL}}\big(q_\phi(\mathbf{z}|\mathbf{x}) \,\|\, q_\phi^{\text{prior}}(\mathbf{z}|\mathbf{x})\big). \tag{5}$$

While the uniform prior adopted in NRI serves as a regularization mechanism to constrain the structure of the latent space, it may significantly deviate from the true underlying distribution. In contrast, IPSI employs a learned prior that functions both as a regularizer and a soft target. This approach not only guides the early stages of training but also mitigates the adverse effect of the KL divergence loss on prediction accuracy during later training phases. Figure 1 shows a comparison between two kinds of priors and the ground truth.

A key challenge in the IPSI framework lies in how to obtain structural labels for supervising the training of $\text{SI}_{\text{prior}}$, since ground-truth structures are unavailable in unsupervised structural inference. In this work, we propose an iterative training framework. In the first round of iteration, the structural labels are derived either from a synthetic meta-dataset or from pseudo-labels inferred by a baseline model, depending on whether prior knowledge of system dynamics is available (more details will be introduced in Section 4.5). In subsequent rounds of iteration, as $\text{SI}_{\text{joint}}$ has been trained and achieves more accurate inference, its outputs are used as new pseudo-labels to update $\text{SI}_{\text{prior}}$. Through this iterative process, both $\text{SI}_{\text{prior}}$ and $\text{SI}_{\text{joint}}$ are progressively refined, this alternating update strategy will help the model escape from local optima and ultimately achieve performance that significantly surpasses that of the baseline models. Figure 2 shows the complete pipeline architecture of IPSI.

Given that IPSI is designed based on the general paradigm of VAE-based structural inference models, it possesses universality and can be integrated into various such models. Next, we will use the NRI model as an example to illustrate how IPSI integrates and enhances the performance of the NRI model.

## 4.2 Pretrained Structure Estimator

The pretrained structural estimator, denoted as $\text{SI}_{\text{prior}}$, adopts the same GNN-based encoder architecture as the joint inference module $\text{SI}_{\text{joint}}$, but is trained in a supervised manner to extract trajectory embedding containing more structural information and generate learnable structural priors, and then input them into the joint training model $\text{SI}_{\text{joint}}$. The encoder comprises a temporal node embedding module followed by a fully connected message-passing network. To better capture the temporal dynamics of each agent's trajectory, we replace the Multi-Layer Perceptron (MLP) embedding used

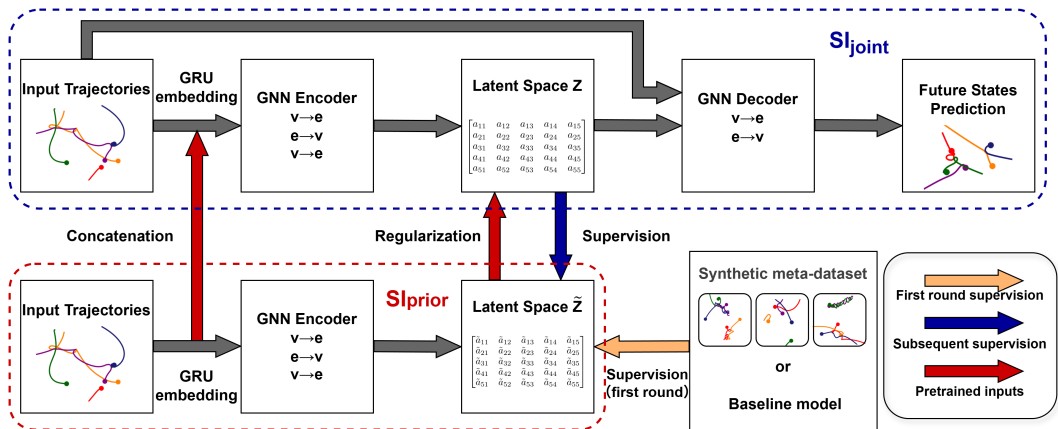

Figure 2: Illustration of the iterative supervision pipeline. The first iteration uses ground-truth labels of meta-dataset or pseudo-labels from baseline model to supervise the training of $\text{SI}_\text{prior}$, which then produces an informative prior to guide $\text{SI}_\text{joint}$. The process is repeated in a loop to progressively improve both modules.

in NRI with a Gated Recurrent Unit (GRU) [3]:

$$\text{Node Embedding:} \quad \mathbf{h}_j^{1,\text{prior}} = \text{GRU}(\mathbf{x}_j), \tag{6}$$

$$v \rightarrow e: \quad \mathbf{h}_{(i,j)}^{1,\text{prior}} = f_e^{\text{prior}}\big([\mathbf{h}_i^{1,\text{prior}}, \mathbf{h}_j^{1,\text{prior}}]\big), \tag{7}$$

$$e \rightarrow v: \quad \mathbf{h}_i^{2,\text{prior}} = f_v^{\text{prior}}\left(\sum_{j \neq i} \mathbf{h}_{(i,j)}^{1,\text{prior}}\right), \tag{8}$$

$$v \rightarrow e: \quad \mathbf{h}_{(i,j)}^{2,\text{prior}} = f_e^{\text{prior}}\big([\mathbf{h}_i^{2,\text{prior}}, \mathbf{h}_j^{2,\text{prior}}]\big). \tag{9}$$

The final edge representation $\mathbf{h}_{(i,j)}^{2,\text{prior}}$ is then used to predict the edge type through a categorical posterior distribution:

$$q_\phi^{\text{prior}}(\mathbf{z}_{ij}|\mathbf{x}) = \text{softmax}\left(\mathbf{h}_{(i,j)}^{2,\text{prior}}\right), \tag{10}$$

where $\phi$ denotes the learnable parameters of the encoder, and $\mathbf{x}$ represents the full input trajectories. We use ground-truth labels $a_{ijk}$ corresponding to $K$ types of edges to supervise the prediction of edge existence via the binary cross-entropy loss:

$$\mathcal{L}_{\text{prior}} = -\sum_{i<j}\sum_{k=1}^{K} a_{ijk} \log q_\phi^{\text{prior}}(z_{ijk} = 1|\mathbf{x}). \tag{11}$$

Details of the source of the structural supervision signal $a_{ijk}$ will be introduced in Section 4.5.

### 4.3 VAE-based Structural Inference Model

The VAE-based structural inference model $\text{SI}_\text{joint}$ adopts an encoder–decoder architecture similar to the NRI framework, but introduces a key enhancement by incorporating prior knowledge from $\text{SI}_\text{prior}$. Specifically, $\text{SI}_\text{joint}$'s encoder concatenates its own node embeddings with those provided by $\text{SI}_\text{prior}$ along the feature dimension. The GRUs in Eq. (6) and Eq. (12) and the learnable prior module in Eq. (18) are independently parameterized, as $\text{SI}_\text{joint}$ and $\text{SI}_\text{prior}$ serve distinct roles: the former includes both an encoder and a decoder for inference and reconstruction, while the latter contains only an encoder that generates structural priors to guide iterative updates. Except for concatenating trajectory embeddings from $\text{SI}_\text{prior}$ after the trajectory embedding layer, the decoder and encoder of

$\mathrm{SI_{joint}}$ are identical to the baseline model (such as NRI):

$$\text{Node Embedding:} \quad \mathbf{h}_j^1 = [\mathrm{GRU}(\mathbf{x}_j), \mathbf{h}_j^{1,\mathrm{prior}}]. \tag{12}$$

It is worth noting that this modification is relatively minor but both effective and widely applicable. Since encoding raw trajectory data into latent node representations is a highly generic step, this design enables IPSI to serve as a general-purpose structural inference enhancement framework that can be flexibly combined with various existing encoder–decoder architectures.

## 4.4   Training with Hybrid Loss

As introduced in Section 4.1, we replace the simple uniform prior $p(\mathbf{z})$ in NRI loss with a learnable prior $q_\phi^{prior}(\mathbf{z}|\mathbf{x})$, which serves both as a regularizer and as a soft target. This learnable prior demonstrates significant advantages over the uniform prior, and, compared to manually crafted structural prior losses proposed in various subsequent works such as sparsity and smoothness, it requires no task-specific design and avoids the difficulty of tuning numerous additional hyperparameters:

$$\mathcal{L} \;=\; \mathcal{L}_p \;+\; \beta \, D_{\mathrm{KL}}\big(q_\phi(\mathbf{z}|\mathbf{x}) \,\|\, q_\phi^{\mathrm{prior}}(\mathbf{z}|\mathbf{x})\big), \tag{13}$$

where $\beta$ controls the influence of the structural prior. Similar to the modifications made in the model, this modification is lightweight yet effective, and can be readily incorporated into most VAE-based structural inference models. As it requires no modification to the base architecture and simply replaces the prior distribution.

## 4.5   Two Sources of Supervision for Pretraining

As introduced in Section 4.1, in order to train $\mathrm{SI_{prior}}$ with structural labels without ground-truth structure of the target system, we introduce two complementary strategies that provide supervision signals under different assumptions about prior knowledge.

### 4.5.1   Supervision from a Synthetic Meta-Dataset with Prior Knowledge

When partial prior knowledge of the system's interaction form is available, we propose constructing a synthetic meta-dataset of labeled interactions. This dataset includes multiple simulated systems with similar dynamical properties but excludes the exact target configuration, aiming to enable $\mathrm{SI_{prior}}$ to learn transferable structural motifs applicable to unseen yet related systems.

For example, in systems governed by radial forces (e.g., springs or charged particles systems), we generate synthetic trajectories from various parameterized systems with edge types such as attractive, repulsive, or null, and with different distance dependencies (e.g., constant, inverse, inverse-square). Importantly, we exclude systems matching the specific parameters of the target system, thereby simulating the following situation: "The observer is aware that interactions are radial in nature, but has no knowledge of the proportion of attractive, repulsive, or null connections, nor the precise functional relationship between force magnitude and distance."

As introduced in Section 4.1, $\mathrm{SI_{prior}}$ is trained on the synthetic meta-dataset using ground-truth structural labels as supervision in the first round of iteration, and trained on the target dataset using pseudo-labels as supervision in the following rounds.

### 4.5.2   Supervision via Pseudo-Labels without Prior Knowledge

When prior knowledge of the system dynamics is unavailable, we first train a baseline model such as NRI and use its structure inference outputs as pseudo-labels to provide supervision. Although this approach does not incorporate the additional information from the synthetic meta-dataset, the pseudo-labels still offer a more informative prior estimate than uniform prior. Furthermore, the subsequent iterative process described in Section 4.1 is applied in the same manner, alternating the optimization of modules $\mathrm{SI_{prior}}$ and $\mathrm{SI_{joint}}$, ultimately leading to improved performance.

# 5 Experiments

## 5.1 Datasets

To evaluate the effectiveness of the two supervision strategies, we focus on the three synthetic physical systems proposed in [8]—the spring, charged particle, and Kuramoto oscillator systems—as our synthetic meta-datasets are constructed based on the spring and charged particle systems. We adopt the same simulation configuration as in the original setup: each system is simulated for 5000 time steps, with 5 or 10 interacting objects, and the interaction graph is set to undirected. The data is split into training, validation, and test sets with a 5:1:1 ratio. And the details about the synthetic meta-dataset are described in supplementary materials.

## 5.2 Baselines

We compare our model with several recent structural inference approaches that focus on physical systems with available and complete codes:

• NRI [8]: a VAE-based structural inference model that jointly learns the relations and dynamics.

• SUGAR [11]: a method that introduces structural prior knowledge for structural inference.

• MPM [2]: a method that combines a relation interaction mechanism and a spatio-temporal message passing mechanism.

• iSIDG [16]: a method that iteratively updates the encoder structure based on the inferred structure.

Our model is evaluated in both the with prior (w/ prior) and without prior (w/o prior) situations, which correspond to different supervision methods as described in Section 4.5. And the evaluation metric is edge classification accuracy, which reflects how accurately the inferred edge types match the ground truth. Due to space limitations, the results on trajectory prediction error are provided in the supplementary materials. We apply IPSI framework on two different models, NRI and MPM, and report the improved performance to evaluate the generality and flexibility of IPSI.

## 5.3 Results

The experimental results are summarized in Table 1. Across three types of physical systems, the performance of models enhanced with IPSI is significantly better than the original model. This improvement is particularly pronounced on the Charged Particle dataset, where the interactions include both attraction and repulsion, leading to more complex motion patterns.

Table 1 also presents the performance of $SI_{prior}$ trained exclusively on the synthetic meta-dataset. While $SI_{prior}$ outperforms the NRI baseline on charged particle datasets, it is still surpassed by the full IPSI pipeline, indicating that $SI_{joint}$ actively refines the initial priors through joint training rather than merely replicating them. Moreover, on datasets with lower baseline accuracy, IPSI performs better when w/ prior, highlighting the complementary benefits of the two supervision strategies. Finally, despite the non-radial nature of interactions in the Kuramoto system, $SI_{prior}$ still provides a reasonable structural estimate, demonstrating its generalization capability. This suggests that constructing a larger synthetic meta-dataset to train a more powerful $SI_{prior}$—inspired by the scaling principles of large language models—could be a promising direction for future work.

## 5.4 Additional Evaluation on the DoSI Benchmark

To further assess the generalization ability of our approach under different graph structures, we conducted additional experiments on the DoSI benchmark [20], which simulates dynamical trajectories over empirically-derived graphs from real-world domains. In particular, we evaluated three biologically inspired datasets, each containing 15 nodes: *Brain Networks (BN)*, *Gene Regulatory Networks (GRN)*, and *Vascular Networks (VN)*. Besides the conventional *Springs* simulator, we additionally used the *NetSims* simulator, which models brain activity by assigning nodes to brain regions and edges to their interactions [14].

For this study, we applied our method under the IPSI (w/o prior) configuration. Table 2 reports the AUROC scores on all six datasets. We observe that IPSI consistently outperforms the baseline methods (NRI [8], MPM [2]) across both simulators, achieving state-of-the-art performance among

Table 1: Edge prediction accuracy (%) on Springs, Charged, and Kuramoto systems. For NRI, SUGAR, and MPM, we directly used the results from [8] and [2] after checking the consistency of the benchmark. For iSIDG [16], we used our own measured results because different datasets generation methods were used.

| Model | Springs | Charged | Kuramoto |
|---|---|---|---|
| *5 objects* | | | |
| NRI | **99.9** $\pm$ 0.0 | 82.1 $\pm$ 0.6 | 96.0 $\pm$ 0.1 |
| SUGAR | **99.9** $\pm$ 0.0 | 82.9 $\pm$ 0.8 | 91.8 $\pm$ 0.1 |
| MPM | **99.9** $\pm$ 0.0 | 93.3 $\pm$ 0.5 | 97.3 $\pm$ 0.2 |
| iSIDG | **99.9** $\pm$ 0.0 | 91.7 $\pm$ 0.6 | 96.9 $\pm$ 0.3 |
| IPSI (NRI-based, w/ prior) | **99.9** $\pm$ 0.0 | 91.2 $\pm$ 0.4 | 97.2 $\pm$ 0.3 |
| IPSI (NRI-based, w/o prior) | **99.9** $\pm$ 0.0 | 89.3 $\pm$ 0.3 | 97.4 $\pm$ 0.3 |
| IPSI (MPM-based, w/ prior) | **99.9** $\pm$ 0.0 | **94.3** $\pm$ 0.4 | 98.0 $\pm$ 0.3 |
| IPSI (MPM-based, w/o prior) | **99.9** $\pm$ 0.0 | **94.3** $\pm$ 0.3 | **98.1** $\pm$ 0.2 |
| $SI_{prior}$ (NRI-based, w/ prior) | 89.7 $\pm$ 0.2 | 86.7 $\pm$ 0.3 | 77.1 $\pm$ 0.5 |
| Supervised | 99.9 $\pm$ 0.0 | 95.4 $\pm$ 0.1 | 99.3 $\pm$ 0.0 |
| *10 objects* | | | |
| NRI | 98.4 $\pm$ 0.0 | 70.8 $\pm$ 0.4 | 75.7 $\pm$ 0.3 |
| SUGAR | 98.3 $\pm$ 0.0 | 72.0 $\pm$ 0.9 | 74.0 $\pm$0.2 |
| MPM | 99.1 $\pm$ 0.0 | 81.6 $\pm$ 0.2 | 80.3 $\pm$ 0.6 |
| iSIDG | 99.2 $\pm$ 0.0 | 82.1 $\pm$ 0.3 | 80.6 $\pm$ 0.5 |
| IPSI (NRI-based, w/ prior) | 99.2 $\pm$ 0.1 | 77.1 $\pm$ 0.5 | 77.3 $\pm$0.4 |
| IPSI (NRI-based, w/o prior) | 99.2 $\pm$ 0.1 | 75.3 $\pm$ 0.3 | 76.9 $\pm$0.3 |
| IPSI (MPM-based, w/ prior) | **99.6** $\pm$ 0.1 | **86.4** $\pm$0.7 | 82.6 $\pm$0.8 |
| IPSI (MPM-based, w/o prior) | **99.6** $\pm$ 0.1 | 86.2 $\pm$0.5 | **82.8** $\pm$0.6 |
| $SI_{prior}$ (NRI-based, w/ prior) | 87.6 $\pm$ 0.4 | 76.2 $\pm$ 0.5 | 73.3 $\pm$ 0.6 |
| Supervised | 99.4 $\pm$ 0.0 | 89.7 $\pm$ 0.1 | 94.3 $\pm$ 0.8 |

VAE-based structural inference models. These results suggest that IPSI remains robust in a more realistic evaluation environment.

## 5.5 Training Dynamics

To further elucidate the training dynamics, we track three metrics during a single run of $SI_{joint}$ training: the state-prediction mean squared error (MSE), the KL-divergence regularization term, and the edge-prediction accuracy on the training set. As illustrated in Figure 3a, the KL term initially decreases—indicating that the model closely follows the pretrained prior—then rises as $SI_{joint}$ diverges to learn more precise structures. In contrast, both MSE and edge accuracy improve steadily throughout training. This behavior suggests that the model first undergoes a phase of prior imitation and subsequently refines the structures based on the initial priors, resulting in enhanced predictive performance. There three experiments are conducted on charged particle dataset with 5 objects and w/o prior situation.

## 5.6 Robustness

Figure 3b illustrates the impact of $SI_{prior}$ accuracy and iteration round on $SI_{joint}$ performance. The results show that even when $SI_{prior}$ is completely untrained (guessing randomly with two edge types will achieve about 50% accuracy), $SI_{joint}$ still achieves the same performance as the NRI baseline. This demonstrates the robustness of the IPSI framework: even when the prior inputs are poor, IPSI is capable of learning to shield against these erroneous priors, maintaining performance comparable to the original baseline model. Meanwhile, the inserted figure shows that the accuracy steadily increases and eventually converges, demonstrating the effectiveness and stability of the proposed iterative training strategy.

Table 2: **Results on the DoSI benchmark (AUROC, %)**. All datasets contain 15 nodes. IPSI (w/o prior) maintains strong performance and achieves the best results among VAE-based approaches. Higher is better.

| Methods | Springs (AUROC) | | | NetSims (AUROC) | | |
|---|---|---|---|---|---|---|
| | BN | GRN | VN | BN | GRN | VN |
| NRI [8] | 99.75 | 90.55 | 92.68 | 99.79 | 78.08 | 89.13 |
| ACD | 99.75 | 91.10 | 94.34 | 99.87 | 80.18 | 80.32 |
| MPM [2] | 99.98 | 94.02 | 96.56 | 99.95 | 76.06 | 91.18 |
| iSIDG | 99.97 | 92.91 | 96.59 | 99.91 | 71.11 | 91.20 |
| RCSI | 99.81 | 93.01 | 97.03 | 99.72 | 77.45 | 91.53 |
| IPSI (NRI-based, w/o prior) | 99.75 | 91.19 | 93.77 | 99.80 | 84.40 | 93.75 |
| IPSI (MPM-based, w/o prior) | **99.98** | **98.95** | **99.39** | **99.97** | **95.32** | **96.75** |

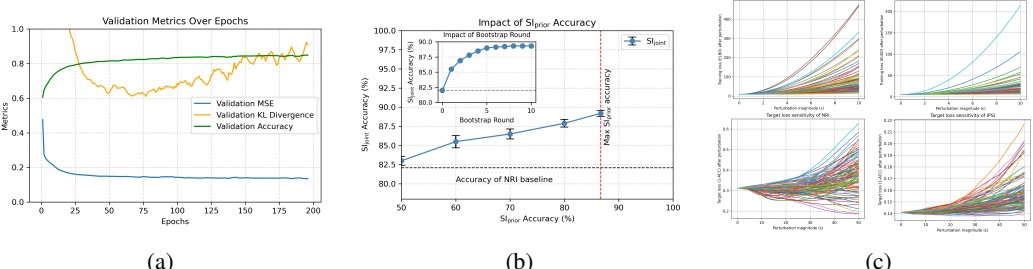

(a)  (b)  (c)

Figure 3: Three additional experiments conducted on the charged particle dataset with 5 objects (a) Training dynamics of $\text{SI}_{\text{joint}}$. (b) Impacts of iteration round and $\text{SI}_{\text{prior}}$ accuracy on $\text{SI}_{\text{joint}}$ accuracy. (c) Loss perturbation analysis of NRI and IPSI. Each curve shows the loss change under random parameter perturbations, where $\varepsilon$ denoting perturbation magnitude (x-axis).

## 5.7 Perturbation Analysis

To gain deeper insight into IPSI, we performed a random-direction perturbation analysis: For both trained NRI and IPSI models, all model parameters were flattened into a single vector, a random unit vector of identical dimension was sampled as a perturbation direction, and scaled perturbations were added to evaluate the change in loss. We analyzed two loss functions: (1) the *training loss* (ELBO objective), and (2) the *target loss* (1-ACC), reflecting structural inference quality. Across 100 random directions, we computed two quantitative measures: *non-optimal proportion* (the fraction of perturbation directions yielding lower loss than the original model, within a tolerance) and *relative improvement* (the mean percentage decrease of loss among those non-optimal directions).

Experiments on the GRN dataset (Springs, 15 nodes, w/o prior) yielded the results summarized in Table 3. Under the training loss, both NRI and IPSI exhibit comparable local optimality, suggesting that the Adam optimizer effectively finds stable basins for both. However, under the target loss, IPSI demonstrates more stable minima, reflected by lower non-optimal proportion and smaller relative improvement. The perturbation line plots in Figure 3c further visualize these differences, where each curve corresponds to a random perturbation direction. For target loss sensitivity of IPSI (bottom right), most curves are monotonically increasing, while target loss sensitivity of NRI (bottom left) shows irregular patterns with many curves decreasing under perturbation. These observations suggest that both models converge to local optima under the training loss, but IPSI converges to better optima under the task-relevant target loss. IPSI effectively **reshapes the loss landscape** through iteratively updates trajectory embeddings and structural priors, thereby aligning the training objective with the structural inference target.

Table 3: Perturbation analysis on GRN (Springs, 15 nodes).

| Model | Training Loss | | Target Loss | |
|---|---|---|---|---|
| | Non-opt. (%) | Rel. Imp. (%) | Non-opt. (%) | Rel. Imp. (%) |
| NRI | 1 | 1.16 | 63 | 31.7 |
| IPSI (w/o prior) | 2 | 1.49 | 6 | 11.2 |

## 5.8 Ablation Study

To analyze the contribution of each pretrained input component, we perform an ablation study on its two injection points within $SI_{joint}$: (1) the *embedding vector* and (2) the *structural prior*. Three reduced variants are evaluated: one using only the embedding vector, one using only the structural prior, and one using neither—equivalent to an NRI variant with a GRU embedding scheme. Experiments are conducted under the w/ prior setting on the charged particle dataset with five objects, and results are shown in Table 4.

Both single-input variants outperform the baseline without pretrained inputs, demonstrating that each component independently improves performance. The full model, combining both the embedding vector and the structural prior, achieves the best accuracy, indicating their complementarity. These findings are consistent with our design rationale: the embedding vector provides enriched representations informed by global structural cues, while the latent prior serves as a regularization signal that steers the latent distribution toward plausible interaction patterns. Together, they enhance the structural inference at both the representation and generative levels.

Table 4: Ablation results on edge accuracy (%) for the Charged dataset with 5 objects.

| Model Variant | Charged Particle System |
|---|---|
| No pretrained input (baseline) | $82.2 \pm 0.4$ |
| Only embedding vector | $89.5 \pm 0.7$ |
| Only structural prior | $84.1 \pm 0.4$ |
| Full model (both inputs) | $\mathbf{91.2} \pm 0.4$ |

## 6 Conclusion and Limitations

In this work, we introduced IPSI, an iterative pretrained structural inference framework that alternates between data-driven inference and structure-guided refinement. Through extensive experiments on both synthetic and empirically derived benchmarks, IPSI consistently improved structural accuracy and generalization over existing VAE-based methods. The perturbation analysis further demonstrated that IPSI enhances the alignment between training and task objectives by iteratively reshaping the loss landscape, providing an intuitive explanation for its superior performance.

A key limitation of IPSI lies in the increased computational overhead introduced by its iterative process, which makes it challenging to evaluate on large-scale systems (involving dozens or more agents) and real-world datasets. Generalizing structural inference to large graphs remains a major challenge, primarily due to the quadratic growth of computational cost with respect to the number of nodes when employing fully connected GNN encoders.

Nevertheless, IPSI offers a promising direction to address this issue. By leveraging the pretrained structural prior to guide the construction of encoder connectivity—rather than using it solely as a latent prior—the effective edge complexity can be reduced from $\mathcal{O}(N^2)$ to $\mathcal{O}(N)$ for systems with sparse underlying structures. In future work, we plan to explore scalable variants of IPSI and apply it to large-scale, real-world dynamical systems such as neural and biological networks, as well as further investigate theoretical properties of its iterative optimization process.

## Acknowledgments

Code is available at `https://github.com/blackbird1729/IPSI`.

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
