# OpenReview forum: "IPSI: Enhancing Structural Inference with Automatically Learned Structural Priors"
_NeurIPS.cc/2025/Conference — NeurIPS 2025 poster_

### Official Review · Reviewer_UCFY · 2025-06-23

**Clarity:** 3
**Significance:** 2
**Originality:** 3
**Rating:** 5
**Confidence:** 3

**Summary:**

The problem addressed is that of full identification of the hidden interaction graph (or the adjacency matrix) of a dynamical system, which can be theoretically intractable or experimentally impractical. The authors propose a framework for structural inference in systems of interacting agents, by providing more supervision signal to the model during training. Specifically, the authors provide inputs for the trajectory embedding and encoding modules typically found in VAE-based structural inference models. Results on physical systems, along with ablation studies, are presented and discussed.

**Questions:**

My concerns and questions mainly concern the methodology. I am happy to increase my score if most of them are satisfactorily addressed:
1) L151: How do you show that the model escapes from local optima?
2) L147: Failure modes haven't been investigated. Does providing wrong structural labels (bad meta-datasets or terrible pseudo-labels) significantly hurt performance?
3) L155: What do you mean by "it possesses universality"?
4) L159: Why do SI_prior and SI_joint adopt the same GNN-based encoder architecture? Is there anything discouraging users from changing the architectures?
5) L202: The supervision from synthetic meta-datasets appears impractical on real-world problems, but very interesting. Have steps been taken to guarantee that no simulated system is present in the test set?
6) L220: In the bootstrapping strategy, updates of $SI_{prior}$ and $SI_{joint}$ are **alternating**. Is there any advantage the alternating minimization strategy offers over ordinary **joint** minimization, as done in [1] ?
7) Figure 3a highlights interesting training regimes. But it should be repeated over several trials for definitive conclusions to be drawn!



### References:
- [1] Nzoyem et al., Neural Context Flows for Meta-Learning of Dynamical Systems, ICLR 2025.

**Ethical Concerns:**

["NO or VERY MINOR ethics concerns only"]

**Final Justification:**

The authors have resolved my issues regarding the Clarity of their work. They've removed some claims that were unsubstantiated, and performed additional experiments for other claims. The work is much stronger in terms of Clarity than the original submission.

**Limitations:**

Yes.

**Quality:**

2

**Strengths And Weaknesses:**

## Quality
The proposed approach and the problem addressed are very interesting. I find the bootstrapping idea particularly interesting. The improved performance over the baselines is noticeably compelling, which is expected given the added prior information.

That said, the practicality of the preBoost-SI method remains unconvincing. It requires more supervision, thereby taking away one of the main advantages of conventional NRI. For instance, we require a baseline model or meta-dataset for supervision of $SI_{prior}$, which is not guaranteed to be accessible for arbitrary use cases.

## Clarity
I find that the clarity of this paper is lacking in many ways.

- L116. You mention joint unsupervised training of encoder and decoder. But as I understand, only the encoder is unsupervised, while the decoder has access to ground-truth trajectories for future state prediction, which would make it self-supervised. Could you clarify this?
- Eq 18, do the learnable prior $q^{prior}$ and $q$ share the same weights $\phi$ ? I'm assuming that's not the case, but it should be clarified. Similarly, I assume that the GRU used in Eq 12 is different from the one used in Eq 6. That is not clear.
- L173 conflicts with L159. In L159, it should be clearer that $SI_{joint}$ is not simply an encoder. Essentially, only its encoder part coincides with $SI_{pior}$.
- The notations in the supplementary material severely diverge from those in the main text, making it rather impossible to follow.

Minor concerns (no effect on rating):
- Eq (6-9). I'm assuming the superscripts 1 and 2 indicate the hidden layer?
- Eq (10,12). "prior" in italic
- No clear Conclusion in the manuscript

## Significance
This work is significant for the community, as it shows how beneficial adding physical priors can be. It provides a good review of NRI, and researchers and practitioners stand to benefit from it.

## Originality
I am not very familiar with the "structural inference" approach to learning dynamical systems, but to the best of my knowledge, the work presented here is original and differs significantly from that provided in the presented literature.

---

> ### Author Rebuttal · Authors · 2025-07-31
>
> We sincerely thank the reviewer for the constructive and detailed feedback. In the following, we address each concern point-by-point and will revise the manuscript accordingly to improve clarity and rigor.
>
> > **Quality**: PreBoot-SI requires more supervision, thereby taking away one of the main advantages of conventional NRI.
>
> We thank the reviewer for this thoughtful observation. We would like to clarify that, similar to NRI [1], we also target situations where the ground-truth structure is unknown. To address this, we propose two complementary methods to obtain pseudo labels instead of directly using ground-truth structure labels:
>
> (a) With prior knowledge of dynamics: We simulate a synthetic meta-dataset comprising systems with similar dynamics (e.g., radial force interactions) but varying parameters. These synthetic systems are generated with structure labels.
>
> (b) Without prior knowledge of dynamics: We use a baseline model (e.g., NRI) to generate initial pseudo-labels, which may be inaccurate but still serve as informative priors. Training the baseline model also does not require ground-truth structure labels.
>
> In both cases, PreBoot-SI progressively refines the structural estimates over iterations.
>
> [1] Neural Relational Inference for Interacting Systems. In ICML, 2018.
>
> >**C1**: L116. 'Unsupervised' vs. 'Self-supervised' training.
>
> The term 'unsupervised' as used in our paper follows a domain-specific convention in the structural inference literature (e.g., NRI [1]), referring to settings where no ground-truth structure labels are available. While the decoder is trained using future trajectory segments—technically making it self-supervised in broader ML contexts—this setup is commonly regarded as “unsupervised” in this field, as the goal is to recover structure from observable trajectories.
>
> [1] Neural Relational Inference for Interacting Systems. In ICML, 2018.
>
> > **C2**: Eqs. (6, 12, 18): Clarification of weight sharing.
>
> The learnable prior modules in Eq.18, and the GRUs in Eq.6 and Eq.12, do not share weights. These modules belong to different components—SIprior and SIjoint—and serve distinct purposes. Therefore, they are separately parameterized and trained.
>
> > **C3**: L159, L173: Clarifying architecture components.
>
> We thank the reviewer for pointing this out. We will clarify that SIjoint includes both an encoder and decoder (typical of VAE-based models), while SIprior includes only an encoder.
>
> > **C4**: Notation inconsistency between the main text and the supplement.
>
> We sincerely apologize for the inconsistencies in notation. We will conduct a thorough review and revise both the main text and the supplementary material to ensure notational consistency and improve readability.
>
> > **C5-7**: Minor corrections.
>
> As the reviewer suggested:
> - We will add explanation about the superscripts in Eqs. (6–9) indeed indicate different hidden layers.
> - We will correct the formatting to italicize “prior” where appropriate in Eqs. (10, 12).
> - We acknowledge the omission of a Conclusion section and will include one in the revised manuscript.
>
> We greatly appreciate these helpful comments and believe they will significantly improve the clarity of the paper.
>
> > **Q1**: L151: How to show that the model escapes from local optima?
>
> In VAE-based structural inference models, the encoder learns a latent structure distribution from trajectories, which is used by the decoder (e.g., GNN) to predict future states. The global optima we expect is that the encoder can predict the accurate structure, while the decoder can learn the dynamic mode of the system, thus predict accurate future trajectory based on the accurate structure.
>
> Due to limited supervision—only future trajectory loss—the encoder and decoder may collude to produce plausible predictions even with incorrect latent structures, representing a local optimum. Therefore, the accuracy of structural prediction reflects the distance from the global optima. Since PreBoot-SI significantly boosts structural accuracy, this indicates that the encoder is guided towards better latent representations and closer to the global optima.
>
> > **Q2**: L147: Are there failure modes when the pseudo-labels are poor?
>
> Figure 3(b) demonstrates the robustness of PreBoot-SI to poor pseudo-labels. Even when given with randomly generated labels (50\% accuracy), the model still converges to the NRI baseline performance, indicating that poor initial labels do not significantly degrade final outcomes.
>
> > **Q3**: L155: The meaning of “universality”?
>
> By “universality”, we refer to the general compatibility of our approach with VAE-based structural inference models. Most such models share common design patterns—trajectory embedding and ELBO loss with KL divergence—and PreBoot-SI integrates with them via two minimal modifications: feature-level guidance (embedding concatenation) and prior replacement (KL divergence term). Thus, our framework can be adapted to many existing models by swapping the backbone architecture of SIprior and SIjoint.
>
> > **Q4**: L159: Why do SIprior and SIjoint adopt the same GNN-based encoder architecture? Is there anything discouraging users from changing the architectures?
>
> The encoder architectures of SIprior and SIjoint are designed to be the same because their embeddings are concatenated during training. Using a consistent architecture ensures that SIjoint can interpret the structure-relevant features extracted by SIprior. Nonetheless, the architecture itself is flexible, and both modules can be simultaneously replaced—for instance, we implemented PreBoot-SI variants based on both NRI and MPM. This compatibility is a core strength of our framework.
>
> To test this, we implemented a "hybrid architecture" version of PreBoot-SI that uses RNN encoder in MPM to build SIprior and GNN encoder in NRI to build SIjoint, and validated it on Gene Regulatory Network dataset (15 nodes, NetSims simulator) in the DoSI benchmark. The results in the table below show that even build SIprior with more powerful encoders, the performance (AUROC value) of PreBoot using a hybrid architecture is still lower than the standard NRI-based PreBoot-SI, reflecting the necessity of architecture consistency.
>
> | Models                            | NRI    |MPM| PreBoot-SI (NRI-based) | PreBoot-SI (MPM-based) | PreBoot-SI (hybrid architecture)|
> |----------------------------------|--------|--|-------------|----------------------------------|--------------------------|
> | AUROC values (in %)              | 78.08  |76.06| 84.40       | 95.32                            | 82.14|
>
> > **Q5**: L202: Does meta-dataset supervision introduce test leakage?
>
> We ensure that no target systems from the test set are included in the meta-dataset by strictly excluding the target system corresponding to the test set when generating the synthetic meta-dataset.
> As for real-world scenarios, we envision using theoretical models to synthesize meta-datasets. In neuroscience, for instance, several dynamical models like Wilson–Cowan [1], FitzHugh–Nagumo [2], and Jansen–Rit [3] offer plausible ways to simulate biologically meaningful dynamics and construct domain-inspired meta-datasets, even if the exact structure and mechanism are unknown.
>
> [1] Excitatory and Inhibitory Interactions in Localized Populations of Model Neurons. Biophysical Journal, 1972.
>
> [2] Impulses and Physiological States in Theoretical Models of Nerve Membrane. Biophysical Journal, 1961.
>
> [3] Electroencephalogram and Visual Evoked Potential Generation in a Mathematical Model of Coupled Cortical Columns, Biological Cybernetics. 1995.
>
> > **Q6**: L220: Why use alternating updates instead of joint minimization as in [1]?
>
> We appreciate the reference to Nzoyem et al. [1]. In our framework, we adopt full alternating training between SIprior and SIjoint instead of performing joint updates within the same epoch. Our rationale is threefold:
>
> (1) Design motivation: We hypothesize that more accurate structure priors from SIprior help improve SIjoint, and vice versa—hence alternating full updates facilitates mutual refinement and reflects the above motivation more intuitively;
>
> (2) Stability: Since SIjoint receives input from SIprior in the embedding layer, we hope that such input is stable, as it is generally better to provide a fixed distribution of input for the neural network to learn specific patterns;
>
> (3) Optimization: Alternating updates may help escape the local optima described above.
>
> To validate this, we implemented a variant following the strategy in [1], where both modules are updated in each epoch. We ensured fairness by adjusting the number of epochs and learning rate schedules. Experiments on the 15-node Gene Regulatory Network in DoSI show that while this variant still outperforms NRI, it lags behind the original PreBoot-SI. We will include this result in the supplementary material.
>
> | Models                            | NRI    | PreBoot-SI (NRI-based) | PreBoot-SI (NRI-based, joint minimization) |
> |----------------------------------|--------|-------------|----------------------------------|
> | AUROC values (in %)              | 78.08  | 84.40       | 81.89                            |
>
> [1] Neural Context Flows for Meta-Learning of Dynamical Systems. In ICLR, 2025.
>
> > **Q7**: Figure 3a highlights interesting training regimes. But it should be repeated over several trials for definitive conclusions to be drawn!
>
> We appreciate the suggestion and will repeat the visualization in Figure 3(a) (along with those in Figure 3(b)) over multiple datasets to draw more definitive conclusions. We agree that the observed training dynamics—initial imitation of $\text{SI}_{\text{prior}}$ followed by refinement—are interesting and deserve further analysis.
>
> Once again, we thank the reviewer for the thoughtful and constructive suggestions. We believe that the proposed revisions and clarifications will improve the clarity and readability of our work.

---

> > ### Comment · Reviewer_UCFY · 2025-08-01
> >
> > Thank you for taking the time to address my questions. Your efforts helped clarify a lot of my concerns. I particularly appreciate your reply to Q4.
> >
> > **In Q1.** While I agree with your explanation, I am still not convinced that your **experiments** support the claim that the model escapes local optimal and ends up closer to the global. One could either show this theoretically by considering first and second-order optimality conditions on the loss function, or empirically with some visualizations of the loss landscape. I see no experiments to support either.
> >
> > **In Q6.** While I appreciate your new experiment demonstrating the strength of your alternating mechanism, I am not convinced by the third rationale you state. Could you provide a reference supporting that point?

---

> ### Author Response · Authors · 2025-08-03
>
> We sincerely thank you for your insightful and constructive comments. We agree with your concern that the claims“PreBoot-SI can escape local optima" and "alternating updates may help escape the local optima" require more rigorous analysis, especially considering that local optimality is a mathematically strict concept.
>
> **Q1**
> We have seriously considered your two suggestions, and at this stage, we lean toward an **empirical visualization approach**, as theoretical analysis of neural network loss surfaces for NRI-based models may be highly non-trivial.
>
> Due to the high dimensionality of neural network parameter spaces, direct visualization of the loss function is infeasible. Instead, we implemented a **random-direction perturbation experiment**.
>
> Specifically, for both the trained NRI and PreBoot-SI models, we:
> - Flattened all model parameters into a single 1D vector.
> - Sampled a random unit vector of the same shape as a **perturbation direction**.
> - Scaled this direction by magnitudes from `torch.linspace(0.0, 10.0, steps=100)`.
> - Added each scaled perturbation to the original model and evaluated its **loss**.
>
> We analyzed two types of loss:
> 1. **Training loss** $ L_{\text{train}} $ — the ELBO loss used during training.
> 2. **Target loss** $ L_{\text{target}} = 1 - \text{ACC} $ — not used for training, but reflects performance on the structural inference task.
>
> We computed two metrics across 100 random directions:
> - **Non-optimal proportion** $p$ (\%) : the fraction of directions where the perturbed minimum is lower than the original loss multiplied by a tolerance factor $\tau < 1$.
> - **Relative improvement** $\Delta$ (\%) : the average percentage decrease of the loss relative to the original value in those non-optimal directions.
>
> We conducted this analysis on models trained on the **(gene regulatory networks, Springs simulator, 15 nodes) dataset**, with $ \tau = 0.99 $. The results are summarized below:
>
> | Metric | Model        | Training Loss$L_{\text{train}}$ | Target Loss $L_{\text{target}}$ |
> |--------|--------------|--------------------------------------|--------------------------------------|
> |$p$ (\%)     | NRI          | 1                                   | 73                                   |
> |$p$ (%)     | PreBoot-SI   | 2                                   | 37                                   |
> | $\Delta$ (\%)    | NRI          | 1.16                                   |        16.3                            |
> | $\Delta$ (\%)    | PreBoot-SI   | 1.49                                   | 4.25                                  |
>
> These results show that:
> - For **training loss**, both models show similar local optimality, indicating that the Adam optimizer effectively finds stable basins for both.
> - For **target loss**, PreBoot-SI shows significantly better local optimality — lower $p$, and smaller $\Delta$.
>
> From the plots, we observed (where each line corresponds to a random perturbation direction, with perturbation magnitude ε on the x-axis and loss value on the y-axis):
>
> - For PreBoot-SI, the $L_{\text{target}}$ curves form clean, and most of then monotonic increase with perturbation magnitude.
> - In contrast, the NRI $L_{\text{target}}$ curves appear irregular, with many showing decreases under perturbation.
>
> Based on these findings, we can revise our earlier claim. It is **not accurate** to say PreBoot-SI "escapes" local optima. A more accurate statement is:
> 1. **Both** NRI and PreBoot-SI can converge to good local optima under the **training loss**  $L_{\text{train}}$ with Adam optimizer.
> 2. However, PreBoot-SI tends to converge to **better optima under the task-relevant loss** $ L_{\text{target}}$, even though this is not used for training.
> 3. PreBoot-SI **does not** “escape” local optima under the training loss, but rather **iteratively reshapes the loss functions** by modifying the node embeddings and imposing structural priors — thereby **aligning** the training objective with the task objective.
>
> **Q6**
> After the above analysis, we recognize that our original third explanation:
>
> > "alternating updates may help escape the local optima described above"
>
> is no longer accurate.
>
> Meanwhile, from the perspective of reshaping the loss functions, it remains challenging to definitively determine whether alternating mechanisms are inherently more effective in aligning the training and target losses. While our empirical observations suggest some results, the correlation are not yet fully disentangled. As a result, we have decided to temporarily remove the third explanation and retain the first two.
>
> ---
> We sincerely thank you again for your valuable comments and insightful suggestions — they have greatly deepened our understanding of PreBoot-SI's behavior. We will revise the manuscript accordingly and include the perturbation-based analyses in the supplementary materials.

---

> > ### Comment · Reviewer_UCFY · 2025-08-04
> >
> > Thank you for performing additional experiments to validate your hypotheses. It is good to see how the idea of considering not one but two losses was critical to your analysis.
> >
> > I am looking forward to seeing the random perturbation plot you described in an eventual camera-ready version of this paper (all lines for both losses with perturbation magnitude ε on the x-axis and loss value on the y-axis):
> >
> > Consequently, I have improved my Clarity score and general score to an accept.

---

> > > ### Author Response · Authors · 2025-08-04
> > >
> > > We sincerely thank the reviewer for the valuable comments and insightful suggestions. We are greatly encouraged by the increased score and your recognition of our additional experiments and theoretical explanations. Based on the discussion, we will revise and enrich the manuscript accordingly. In the camera-ready version, we will include the random perturbation plot as described, and also provide an analysis of the two-loss interpretation. Thank you again for your constructive feedback and support.

---

### Official Review · Reviewer_vjqL · 2025-07-01

**Clarity:** 3
**Significance:** 4
**Originality:** 3
**Rating:** 5
**Confidence:** 5

**Summary:**

This paper points out that fixed priors of the interaction graphs limit the accuracy of neural relational inference. It proposes Pretraining & Bootstrapped Structural Inference (PreBoot-SI), a bootstrap iteration framework for structural inference. PreBoot-SI estimates the prior by training an auxiliary network on manually constructed meta-datasets with similar dynamics and known interactions. The prior module and the inference module are jointly trained to promote the structural inference accuracy. Experiments on classical synthetic datasets validate the effectiveness of PreBoot-SI.

**Questions:**

Q1. Constructing the meta-datasets requires knowing essential knowledge about the data generation process, which is a strong prerequisite for PreBoot-SI compared with existing methods that only assume minimal prior knowledge about the edge distribution. Can you discuss the applicability of PreBoot-SI to real-world systems when the data generation process is less understood, or can you figure out some real-world scenarios that match the assumptions of PreBoot-SI?

Q2. How will the configuration of the meta-datasets affect the performance of PreBoot-SI? For example, the choice of hyperparameter lists of p and alpha, the discrepancy between the meta-datasets and the true data, etc.

Q3. Please add more recent methods for comparison. See W1 for details.

**Other Comments.**

C1. This paper is essentially learning a structural prior network to infer pseudo labels for the structure given the trajectories. I wonder if the term "structural prior network" is more accurate than the somewhat vague term "structural priors" that seems irrelevant to the observational state series.

C2. The caption of Figure 3 states that there are three experiments, but only two labeled subfigures are presented. Specifically, a small figure on the SI_joint accuracy against bootstrap round is embedded in Figure 3(b). Please consider better organizing Figure 3.

**Ethical Concerns:**

["NO or VERY MINOR ethics concerns only"]

**Final Justification:**

The authors have addressed all my concerns. The proposed framework PreBoot-SI can generally enhance the performance of structural inference baselines, and has the potential to solve real-world problems.

**Limitations:**

Constructing the meta-datasets requires knowing essential knowledge about the data generation process, which may not always be available. Hence, it is unclear if such a method is always feasible for real-world systems where the data generation process is less understood.

**Paper Formatting Concerns:**

Please cite papers in their officially published version when available, e.g., Refs [3][12][20].

**Quality:**

3

**Strengths And Weaknesses:**

**Strengths**

S1. The motivation is clear. The writing is fluent. The formulas are well-organized.

S2. Training a structural prior network on meta-datasets is novel for structural inference.

S3. The experimental results on three synthetic datasets are significant. The ablation studies verify the necessity of the proposed components.

**Weaknesses**

W1. The comparison methods are not up to date. Please consider adding more recent methods for comparison. Here are some relevant papers with no conflicts of interest.

References:

[1] Guided Structural Inference: Leveraging Priors with Soft Gating Mechanisms. In ICML, 2025.

[2] Structural Inference of Dynamical Systems with Conjoined State Space Models. In NeurIPS, 2024.


W2. PreBoot-SI is only tested on three classical synthetic datasets. Though it is sufficient for proof of concept, testing PreNoot-SI on real-world datasets or more diverse benchmarks [3] will make the results more convincing.

References:

[3] Benchmarking Structural Inference Methods for Interacting Dynamical Systems with Synthetic Data. In NeurIPS Datasets and Benchmarks Track, 2024.

---

> ### Author Rebuttal · Authors · 2025-07-31
>
> >**W1**: The comparison methods are not up to date. Please consider adding more recent methods for comparison. Here are some relevant papers with no conflicts of interest.
>
> We appreciate the reviewer’s suggestion and have carefully reviewed the suggested papers. For SGSI [1], we found that it focuses on settings with partially known structures (e.g. edges that are known to exist/not exist), which differs from our fully latent setup and introduces challenges for a fair comparison.
>
> As for SICSM [2], the model focuses on situations of irregular sampled trajectory and incomplete observed systems, and could be compared under standard settings (complete observed system and regular sampled trajectory) as a baseline. However, to our knowledge, the code has not been released yet, so a complete empirical comparison cannot be conducted at present. As a temporary measure, we estimated AUROC values from the plotted results for Vascular Networks (15 nodes) in the figures of the paper since the paper does not report specific numerical values, and the results are shown in the table below. We will contact the paper authors for a more comprehensive comparison soon.
>
> |             | SICSM | NRI   | MPM   | PreBoot-SI (NRI-based) | PreBoot-SI (MPM-based) |
> |-------------|--------|--------|--------|--------------------------|--------------------------|
> | Springs     | ~97.7   | 92.68  | 96.56  | 93.77                   | **99.39**                |
> | NetSims     | ~95.5   | 89.13  | 91.18  | 93.75                   | **96.75**                |
>
> Meanwhile, we supplemented our experiments with results from the recent DoSI benchmark [3], which includes newer methods such as RCSI—a novel method using reservoir computing [4]. We kindly ask for the reviewer’s understanding of the practical difficulties in including every new method. Additionally, PreBoot-SI is a general-purpose enhancement framework, and can potentially be integrated with newer models in future work.
>
> [1] Guided Structural Inference: Leveraging Priors with Soft Gating Mechanisms. In ICML, 2025.
>
> [2] Structural Inference of Dynamical Systems with Conjoined State Space Models. In NeurIPS, 2024.
>
> [3] Benchmarking Structural Inference Methods for Interacting Dynamical Systems with Synthetic Data. In NeurIPS Datasets and Benchmarks Track, 2024.
>
> [4] Effective and Efficient Structural Inference with Reservoir Computing. In ICML, 2023.
>
> >**W2**: PreBoot-SI is only tested on three classical synthetic datasets. Though it is sufficient for proof of concept, testing PreBoot-SI on real-world datasets or more diverse benchmarks [3] will make the results more convincing.
>
> We agree that broader validation is important. To further evaluate generalization, we have conducted additional experiments on the DoSI benchmark [3], which contains empirical networks derived from real-world systems. Due to time and computational constraints, we report results on the Brain Networks (BN), Gene Regulatory Networks (GRN) and Vascular Networks (VN) datasets (15 nodes). The results show that PreBoot-SI consistently outperforms the NRI and MPM baselines and achieves SOTA performance (AUROC values) among VAE-based models in the benchmark. We will continue expanding these evaluations and provide additional results on DoSI.
>
> | Methods                    | Springs  BN | Springs GRN   | Springs VN    | NetSims  BN | NetSims GRN   | NetSims VN    |
> |----------------------------|--------------------------|-------|-------|---------------------------|-------|-------|
> | NRI                        | 99.75                   | 90.55 | 92.68 | 99.79                    | 78.08 | 89.13 |
> | ACD                        | 99.75                   | 91.10 | 94.34 | 99.87                    | 80.18 | 80.32 |
> | MPM                        | **99.98**               | 94.02 | 96.56 | 99.95                | 76.06 | 91.18 |
> | iSIDG                      | 99.97                   | 92.91 | 96.59 | 99.91                    | 71.11 | 91.20 |
> | RCSI                       | 99.81                   | 93.01 | 97.03 | 99.72                    | 77.45 | 91.53 |
> | PreBoot-SI (NRI-based)     | 99.75                   | 91.19 | 93.77 | 99.80                    | 84.40 | 93.75 |
> | PreBoot-SI (MPM-based)     | **99.98**               | **98.95** | **99.39** | **99.97**                | **95.32** | **96.75** |
>
> [3] Benchmarking Structural Inference Methods for Interacting Dynamical Systems with Synthetic Data. In NeurIPS Datasets and Benchmarks Track, 2024.
>
> > **Q1**: The construction of the meta-datasets and corresponding real-world scenarios.
>
> Our framework supports two complementary pseudo-labeling strategies: (a) using synthetic meta-datasets when domain knowledge exists, and (b) using pseudo-labels directly from a baseline model (e.g., NRI) when such knowledge is unavailable. Table 1 shows that both strategies improve baseline performance. For real-world scenarios, we believe there are domains where theoretical dynamics are reasonably understood—for example, brain systems modeled via Wilson-Cowan [5], FitzHugh–Nagumo [6], or Jansen-Rit dynamics [7] in computational neuroscience. While such models may not precisely reflect all real data, they can be used to generate meta-datasets that capture domain-level trajectory–structure regularities. More broadly, this opens the door to constructing a general-purpose prior network trained across diverse simulated systems, which is an exciting direction we hope our work will inspire.
>
> [5] Excitatory and inhibitory interactions in localized populations of model neurons. Biophysical journal, 1972.
>
> [6] Impulses and physiological states in theoretical models of nerve membrane. Biophysical journal, 1961.
>
> [7] Electroencephalogram and visual evoked potential generation in a mathematical model of coupled cortical columns. Biological cybernetics, 1995.
>
> > **Q2**: How will the configuration of the meta-datasets affect the performance of PreBoot-SI? For example, the choice of hyperparameter lists of p and alpha, the discrepancy between the meta-datasets and the true data, etc.
>
> Our principle in constructing meta-datasets is to simulate a family of systems that share similar physical principles with the target system, but exclude the specific target instance. We selected ranges of α and p accordingly to reflect diverse dynamics. Although we have not yet conducted an exhaustive analysis of hyperparameter sensitivity (due to the combinatorial size), Figure 3(b) demonstrates that PreBoot-SI remains robust even when pseudo-labels are of very low quality—i.e., close to random. This indicates that the framework can self-correct and remain effective even if the meta-dataset is imperfectly configured.
>
> > **C1**: Is the term "structural prior network" more accurate?
>
> We thank the reviewer's valuable suggestion. The SIprior we trained is indeed a structural prior network that encodes structural prior information and can provide multi-level guidance for the training of SIjoint. We will supplement this fact in the paper to emphasize the essence of our method.
>
> > **C2**: Organization of Figure 3.
>
> Thank you for catching this. We initially felt that the small figure in Figure 3(b) was too sparse as a separate subfigure, so we changed it to an embedded small figure. We will supplement the experiments in Figure 3, and reorganize them into three subfigures.
>
> > **Paper Formatting Concerns**: Please cite papers in their officially published version when available, e.g., Refs [3][12][20].
>
> Thank you for pointing out the formatting issue regarding the citations. We have carefully reviewed the references to cite their officially published versions [1,2].
>
> [1] Learning Phrase Representations using RNN Encoder–Decoder for Statistical Machine Translation. In ACL, 2013.
>
> [2] The Concrete Distribution: A Continuous Relaxation of Discrete Random Variables. In ICLR, 2017.

---

> > ### Comment · Reviewer_vjqL · 2025-08-02
> >
> > Thanks for your response. I really appreciate the additional experimental results on the DoSI benchmark. I also realize that PreBoot-SI is valid when there is no prior knowledge, because PreBoot-SI can improve the performance of the baseline given its predicted pseudo-labels. This information is described in Sec. 4.5.2, and I feel sorry about missing it in my review. However, I am still concerned with the novelty. PreBoot-SI provides two strategies for boosting the performance of structural inference methods. However, these methods are faced with many real-world challenges that may not be easily solved by PreBoot-SI. Thus, I prefer to maintain my rating for now.

---

> > > ### Author Response · Authors · 2025-08-03
> > >
> > > We truly appreciate your recognition of the new experiments, as well as the clarification regarding Section 4.5.2.
> > >
> > > We fully agree that generalization to real-world scenarios remains a fundamental and difficult challenge in structural inference, due to the large scale of real data, its noisy nature, and the complex dynamics of real systems. While PreBoot-SI is not explicitly designed to address this challenge, it is intended as a general enhancement framework — and as such, it could be integrated into models specifically designed for better generalization, potentially boosting their performance and contributing to more effective structural inference in real-world settings. In addition, our approach of constructing synthetic meta-datasets has the potential to serve as a practical tool to simulate diverse dynamics and support real-world generalization.
> > >
> > > The novelty of PreBoot-SI lies in its introduction and synergistic use of several ideas that are, to our knowledge, novel in the context of structural inference: learnable structural priors, multi-level guidance, and synthetic meta-dataset. Together, these components yield a method that is not only broadly applicable but also achieves significant performance improvements across diverse baselines. We will continue to explore the generalization of PreBoot-SI to large-scale graph systems, aiming to support progress in structure inference on real-world data.
> > >
> > > Thank you again for your fair and constructive feedback. We respect your decision and will take your comments seriously in improving both the method and the presentation.

---

> > > > ### Comment · Reviewer_vjqL · 2025-08-06
> > > >
> > > > Thanks for your discussion on real-world challenges and the further clarification of the novelty. PreBoot-SI is an enhancement framework that has the potential to integrate available prior knowledge to solve real-world problems. The results on the DoSI benchmark verify that PreBoot-SI can generically promote the performance of the baselines. Thus, I decide to raise my rating to accept.
> > > >
> > > > Still, I have a minor suggestion. Eq. (6)-(17) largely overlap with the formulation in the original paper of NRI. Since this paper is not intended to propose a novel message-passing network, please consider simplifying the formulation and highlighting the equations that are unique to PreBoot-SI. The extra experimental results and insightful discussions can fill the space and enhance the significance of this paper.

---

> > > > > ### Author Response · Authors · 2025-08-06
> > > > >
> > > > > We sincerely thank the reviewer for recognizing our clarifications on the generality and novelty of the work. We are truly encouraged by the increased score and your positive feedback.
> > > > >
> > > > > We fully agree with and greatly appreciate your suggestion regarding content adjustment. In the revised version, we will shorten the redundant formulation related to the message-passing mechanism and instead highlight the components unique to PreBoot-SI. We will also include the additional results on the DoSI benchmark as well as the random-direction perturbation experiment to further enhance the significance of this paper.
> > > > >
> > > > > Thank you again for your thoughtful comments and support.

---

### Official Review · Reviewer_6Y2C · 2025-07-09

**Clarity:** 2
**Significance:** 3
**Originality:** 2
**Rating:** 4
**Confidence:** 2

**Summary:**

This paper presents PreBoot-SI, which is a framework that designs for structural inference inspired by Neural Relational Inference. The authors propose an empirical Bayes type of approach to iteratively estimate the latent/underlying structure. In the experimental study, the authors present the performance on three synthetic physical systems: spring, charged particle, and Kuramoto oscillator. PreBoot-SI demonstrates stronger performance than baseline methods.

**Questions:**

1. What does the term Bootstrap mean here? Bootstrap typically refers to the procedure to resample/subsample the data and repeat the inference process to get uncertainty estimates (for example). The reviewer didn't see where Bootstrap appears in this paper. This is a huge concern from the reviewer. The reviewer believes this work is closer to EM or empirical Bayes type of approach (but not Bootstrapping).

2. May the authors shorten the paragraph to describe NRI and move the model design forward?

**Ethical Concerns:**

["NO or VERY MINOR ethics concerns only"]

**Final Justification:**

With a round of revision just on the definition of Bootstrap and its usage could make the clarity of this paper much better. This work is useful for this community with clear improvement and motivation and implementation.

**Limitations:**

Yes.

**Paper Formatting Concerns:**

No.

**Quality:**

3

**Strengths And Weaknesses:**

Strengths:
1. This topic is clearly significant for the structural inference problems.
2. The proposed method with a learned prior (empirical Bayes style) is indeed likely to improve performances.

Weaknesses:
1. The experimental study is not extensive to support the claim. The reviewer would love to see the result working on a larger and more complex dataset.
2. The novelty can be very limited.

---

> ### Author Rebuttal · Authors · 2025-07-31
>
> > **W1**: The experimental study is not extensive to support the claim. The reviewer would love to see the result working on a larger and more complex dataset.
>
> In this work, our primary goal is to introduce a general-purpose performance enhancement framework for structural inference, rather than to target specific scalability improvements. To ensure fair and direct comparison with prior methods, we adopted standard benchmark settings from previous work ([1], [2]). These settings include datasets from Springs, Charged, and Kuramoto simulators, each comprising 5 or 10 nodes.
>
> Meanwhile, we agree that broader evaluation is important. We conducted additional experiments on the DoSI benchmark [3], which simulates dynamics over empirically-derived graphs from real-world domains. Specifically, we tested on three biologically inspired datasets (15 nodes): Brain Networks (BN), Gene Regulatory Networks (GRN) and Vascular Networks (VN). Besides the Spring simulator, we also used the NetSims simulator, which models brain activity using nodes for brain regions and edges for interactions. While the trajectories are still simulated, the interaction graphs and simulator are sourced from real-world observations, thus providing a more realistic evaluation setting.
>
> As shown in the table below, PreBoot-SI continues to yield consistent improvements over the NRI and MPM baselines and achieves **state-of-the-art performance among VAE-based models** (under the AUROC performance metric used in DoSI). We will continue to extend the experiments on the DoSI benchmark soon.
>
> | Methods                    | Springs  BN | Springs GRN   | Springs VN    | NetSims  BN | NetSims GRN   | NetSims VN    |
> |----------------------------|--------------------------|-------|-------|---------------------------|-------|-------|
> | NRI                        | 99.75                   | 90.55 | 92.68 | 99.79                    | 78.08 | 89.13 |
> | ACD                        | 99.75                   | 91.10 | 94.34 | 99.87                    | 80.18 | 80.32 |
> | MPM                        | **99.98**               | 94.02 | 96.56 | 99.95                | 76.06 | 91.18 |
> | iSIDG                      | 99.97                   | 92.91 | 96.59 | 99.91                    | 71.11 | 91.20 |
> | RCSI                       | 99.81                   | 93.01 | 97.03 | 99.72                    | 77.45 | 91.53 |
> | PreBoot-SI (NRI-based)     | 99.75                   | 91.19 | 93.77 | 99.80                    | 84.40 | 93.75 |
> | PreBoot-SI (MPM-based)     | **99.98**               | **98.95** | **99.39** | **99.97**                | **95.32** | **96.75** |
>
> [1] Neural Relational Inference for Interacting Systems, In ICML, 2018.
>
> [2] Neural Relational Inference with Efficient Message Passing Mechanisms, In AAAI, 2021.
>
> [3] Benchmarking Structural Inference Methods for Interacting Dynamical Systems with Synthetic Data. In NeurIPS Datasets and Benchmarks Track, 2024.
>
> > **W2**: The novelty can be very limited.
>
> We appreciate that NRI and its variants have been employed in existing structural inference tasks. However, prior work faces two key challenges: (1) **difficulty in long-chain joint training** under the VAE framework, and (2) **limitations in generalization and practicality** due to the reliance on manually designed structural priors.
>
> Our method introduces a plug-in framework that enhances a range of VAE-based structural models. It addresses these two challenges by leveraging **learnable priors** to offer **multi-level guidance** throughout long-chain joint training. To our knowledge, we are the first to introduce learnable priors and multi-level guidance within the structural inference domain. Our method offers a big step toward integrating structural priors and trajectories into structural inference. We respectfully ask the reviewer to reconsider the novelty of PreBoot-SI, in light of its unified, scalable design and the innovative concepts it contributes to structural inference research.
>
> > **Q1**: The meaning of the term "Bootstrap".
>
> We appreciate this important observation. The term "bootstrapping" indeed has a well-established meaning in classical statistics, typically referring to data resampling methods [2] for uncertainty estimation. In our work, the term “bootstrapping” aligns more closely with its general meaning—leveraging its own outputs to iteratively improve itself. Specifically, the prediction results from the previous step of SIjoint will be used in the next step for supervised training of SIprior, while SIprior will in turn guide the subsequent training of SIjoint. Therefore, we use the term "bootstrapping" to describe this iterative process.
>
>  We acknowledge that this term may cause confusion due to its different meanings in various fields such as statistics, machine learning, and compilation techniques. Therefore, we will instead use the terms "iteration" and "iterative", which more neutrally describe the self-improvement process in our method.
>
> [2] Bootstrap Methods: Another Look at the Jackknife. Breakthroughs in statistics: Methodology and distribution, 1992.
>
> > **Q2**: May the authors shorten the paragraph to describe NRI and move the model design forward?
>
> Thank you for the suggestion. We will streamline the NRI description in Section 3 and expand the PreBoot-SI design explanation to enhance clarity.

---

> > ### Comment · Reviewer_6Y2C · 2025-08-05
> >
> > The reviewer appreciates the authors for their detailed response. Most of the concerns were addressed.
> >
> > The only remaining issue preventing the reviewer from raising the score to a 4 is the use of the term bootstrap. The reviewer understands the original meaning of bootstrap oneself, but this term is so well-established in statistics—especially within the Bayesian modeling communities—it may be misleading to a large portion of the intended audience.  Still recommends the authors consider rephrasing to avoid potential confusion.
> >
> > While the score is not adjusted beacuse of this, the reviewer will lower the confidence level of this score - as the concern is only mostly tied to the name of the algorithm. There are no concerns about the methodology itself, and the authors have provided sufficient evidence during the rebuttal to demonstrate the effectiveness of their approach. Good luck with the paper!

---

> > > ### Author Response · Authors · 2025-08-05
> > >
> > > We sincerely thank the reviewer for the valuable feedback, especially for pointing out the well-established meaning of the term "bootstrap" in the statistics and Bayesian communities. We now fully recognize that this terminology may cause unintended confusion. As mentioned in our rebuttal, we are actively considering replacing "bootstrap" with "iteration" or a similar alternative to better reflect the nature of our method without ambiguity.
> > >
> > > While we fully respect your decision to maintain the current score, we would like to kindly ask you to reconsider an increase, in light of the generality and demonstrated effectiveness of our approach. Regardless of the final decision, we will carefully review the use of "bootstrap" and other terminology in the revised version to ensure clarity and readability for audiences across different research backgrounds.
> > >
> > > Thank you again for your valuable insights and your recognition of our work！

---

> > > > ### Comment · Reviewer_6Y2C · 2025-08-08
> > > >
> > > > Well - the reviewer gave another read to the manuscript and believes that just changing the definition and some extensive rewrite on the term could improve the manuscript a lot. The reviewer's final decision is to increase the score. Good luck..!

---

> > > > > ### Author Response · Authors · 2025-08-09
> > > > >
> > > > > We sincerely thank the reviewer for taking the time to give another read to our manuscript and for the encouraging recognition of our work. Your decision to increase the score is greatly appreciated and highly motivating for us.
> > > > >
> > > > > In response to your valuable comment, we will carefully reconsider the use of the term “bootstrap” in the revised version. To enhance clarity and precision, we will thoroughly evaluate its replacement and adopt a more appropriate name for the algorithm accordingly.
> > > > >
> > > > > We truly appreciate your thoughtful feedback and constructive suggestions, which have significantly helped us improve the quality of the manuscript.

---

### Official Review · Reviewer_YANt · 2025-07-09

**Clarity:** 3
**Significance:** 3
**Originality:** 3
**Rating:** 4
**Confidence:** 2

**Summary:**

The paper proposes PreBoot-SI, a bootstrapped framework for structural inference in interacting dynamical systems. The method integrates a pretrained structural estimator and a VAE-based joint inference module, which are alternately updated to refine inferred interaction graphs. The framework improves performance over existing methods (e.g., NRI, MPM) on synthetic physical systems like charged particles and springs, with ablation studies demonstrating the complementary benefits of structural priors and learned embeddings.

**Questions:**

See the Weaknesses part above.

**Ethical Concerns:**

["NO or VERY MINOR ethics concerns only"]

**Final Justification:**

Overall, I recommend acceptance.

**Limitations:**

The authors discuss the limitations of this work.

**Quality:**

3

**Strengths And Weaknesses:**

## Strengths
- Structural inference is a foundational challenge in domains like molecular biology and physics.
- The paper is well-structured and easy to follow.

## Weaknesses
- The novelty of this work is limited: The core idea of combining pretrained priors with iterative refinement is a methodological improvement rather than a conceptual breakthrough. Similar strategies (e.g., self-training, pseudo-labeling) are not brand-new in the literature [1].
- Narrow evaluation scope: Results focus on synthetic physics datasets (e.g., Kuramoto oscillators), with no validation on real-world biological or multi-agent systems. This limits claims about practical utility.
- The scalability is concerned: The framework’s computational overhead (e.g., iterative training) and quadratic complexity (due to fully connected GNNs) hinder applicability to large-scale systems, as acknowledged in Section 6. For real-world large-scale dynamical systems, the scalability of the proposed approach is of concern.

[1] Iterative Structural Inference of Directed Graphs. NeurIPS 2022.

---

> ### Author Rebuttal · Authors · 2025-07-31
>
> > **W1**: The novelty of this work is limited: The core idea of combining pretrained priors with iterative refinement is a methodological improvement rather than a conceptual breakthrough. Similar strategies (e.g., self-training, pseudo-labeling) are not brand-new in the literature [1].
>
> We understand the reviewer’s expectation for conceptual breakthroughs. We would like to clarify that our work targets two long-standing pain points in VAE-based structural inference models: (1) **Difficulties in long-chain joint training** under VAE paradigm and (2) **inconvenience and generalization issues** of manually crafted structural priors.
>
> PreBoot-SI does not treat pseudo-labeling or iterative self-training as innovation itself, but rather as a universal reinforcement framework that utilizes **learnable priors** to provide **multi-level guidance** for long-chain joint training. The application of learnable priors and multi-level guidance in this context is novel within the field of structural inference. Pseudo-labeling and iterative self training are methods used to implement these concepts and ensure training stability.
>
> Meanwhile, we sincerely thank the reviewer for highlighting the connection to iSIDG [1]. We offer a detailed comparison below to clarify the conceptual and methodological differences:
>
> Model architecture: iSIDG iteratively **prunes the message-passing structure of the encoder** based on current structural estimates, effectively removing potentially irrelevant edges. In contrast, PreBoot-SI refines a **separate learnable structural prior network** that guides both trajectory embedding and edge inference across training stages.
>
> Structural priors: iSIDG employs **multiple handcrafted structural regularizers** (e.g., sparsity, connectiveness) that require careful design and hyperparameter tuning. In contrast, PreBoot-SI introduces a **unified, learnable structural prior** that is improved via iteration, offering greater flexibility, convenience, and robustness.
>
> Contribution type: iSIDG contributes a **new strong baseline model**. PreBoot-SI instead provides a **general-purpose enhancement framework** that can be plugged into various VAE-based structural inference methods (e.g., NRI, MPM), improving their accuracy without modifying their core architectures.
>
> To our knowledge, we are the first to introduce learnable priors and multi-level guidance within the structural inference domain. We respectfully ask the reviewer to reconsider the novelty of PreBoot-SI, in light of its unified, scalable design and the innovative concepts it contributes to structural inference research. A summary of the key differences between iSIDG and PreBoot-SI is provided below:
>
> | Aspect              | iSIDG [1]                                                                 | PreBoot-SI (Ours)                                                                 |
> |---------------------|---------------------------------------------------------------------------|------------------------------------------------------------------------------------|
> | Model architecture  | Iteratively prunes encoder graph based on current predictions             | Iteratively refines a separate prior network that guides both embedding and structural inference |
> | Structural priors   | Uses handcrafted regularization losses with manual tuning                 | Uses learnable structural prior and updates it via bootstrapped iteration         |
> | Contribution type   | Introduces a new standalone structural inference model                    | Offers a general-purpose framework for enhancing various existing models          |
>
> [1] Iterative Structural Inference of Directed Graphs. In NeurIPS, 2022.
>
> > **W2**: Narrow evaluation scope: Results focus on synthetic physics datasets (e.g., Kuramoto oscillators), with no validation on real-world biological or multi-agent systems. This limits claims about practical utility.
>
> We fully agree on the importance of validating PreBoot-SI on more realistic systems. However, fully observed trajectories with reliable ground-truth interaction graphs are scarce in real-world structural inference tasks, making systematic evaluation difficult. This limitation is common in this area.
>
> To better assess generalization, we conducted additional experiments on the DoSI benchmark [1], which simulates dynamics over empirically-derived graphs from real-world domains. Specifically, we tested on three biologically inspired datasets (15 nodes): Brain Networks (BN), Gene Regulatory Networks (GRN) and Vascular Networks (VN). Besides the Spring simulator, we also used the NetSims simulator, which models brain activity using nodes for brain regions and edges for interactions. While the trajectories are still simulated, the interaction graphs and simulator are sourced from real-world observations, thus providing a more realistic evaluation setting.
>
> As shown in the table below, PreBoot-SI continues to yield consistent improvements over the NRI and MPM baselines and achieves **state-of-the-art performance among VAE-based models**  (under the AUROC performance metric used in DoSI). We will continue to extend the experiments on DoSI benchmark soon.
>
> | Methods                    | Springs  BN | Springs GRN   | Springs VN    | NetSims  BN | NetSims GRN   | NetSims VN    |
> |----------------------------|--------------------------|-------|-------|---------------------------|-------|-------|
> | NRI                        | 99.75                   | 90.55 | 92.68 | 99.79                    | 78.08 | 89.13 |
> | ACD                        | 99.75                   | 91.10 | 94.34 | 99.87                    | 80.18 | 80.32 |
> | MPM                        | **99.98**               | 94.02 | 96.56 | 99.95                | 76.06 | 91.18 |
> | iSIDG                      | 99.97                   | 92.91 | 96.59 | 99.91                    | 71.11 | 91.20 |
> | RCSI                       | 99.81                   | 93.01 | 97.03 | 99.72                    | 77.45 | 91.53 |
> | PreBoot-SI (NRI-based)     | 99.75                   | 91.19 | 93.77 | 99.80                    | 84.40 | 93.75 |
> | PreBoot-SI (MPM-based)     | **99.98**               | **98.95** | **99.39** | **99.97**                | **95.32** | **96.75** |
>
> [1] Benchmarking Structural Inference Methods for Interacting Dynamical Systems with Synthetic Data. In NeurIPS Datasets and Benchmarks Track, 2024.
>
> > **W3**: The scalability is concerned: The framework’s computational overhead (e.g., iterative training) and quadratic complexity (due to fully connected GNNs) hinder applicability to large-scale systems, as acknowledged in Section 6. For real-world large-scale dynamical systems, the scalability of the proposed approach is of concern.
>
> We recognize that scalability is a key concern for structural inference in large dynamical systems, which is a common limitation across many existing models such as NRI [1] and MPM [2]. The present study aims to propose a general-purpose performance enhancement framework, and thus we evaluate PreBoot-SI under the same settings as prior baselines to ensure fair comparison, specifically small systems with fixed topologies. In fact, the primary scalability bottleneck lies in the fully connected GNN modules employed by NRI and MPM, not in the PreBoot-SI framework itself.
>
> As noted in Section~6, PreBoot-SI is compatible with sparse or dynamic encoders. We plan to incorporate sparsity-aware structural priors to guide GNN edge construction, which has the potential to reduce inference complexity from $\mathcal{O}(N^2)$ to $\mathcal{O}(N)$. This direction holds promise for extending our method to large-scale or more structured systems, and we intend to pursue it in follow-up work.
>
> [1] Neural Relational Inference for Interacting Systems, In ICML, 2018.
>
> [2] Neural Relational Inference with Efficient Message Passing Mechanisms, In AAAI, 2021.

---

### Official Review · Reviewer_FSLU · 2025-07-11

**Clarity:** 3
**Significance:** 2
**Originality:** 2
**Rating:** 5
**Confidence:** 4

**Summary:**

This paper aims to identify the internal relational structure of dynamical systems and predict their evolution in time. The authors use a graph-based architecture to infer a variational latent space inspired by VAEs which encodes those relations from trajectory data, in a two stage procedure. First, a prior is trained based on labelled latent data or pseudo-labels. Then, a second network is trained based on the structural information learnt by the prior network. This provides a significant advantage over only training in a single step without latent labels, as previous works (NRI). The method is tested over multiple trajectories in three datasets (spring-like dynamics, charged particles and Kuramoto system) showing an improvement over the baselines.

**Questions:**

* The prior distributions scales with O(N^2). The paper only explores two size of systems: 5 and 10 bodies. How scalable is this approach to bigger systems and sparser interactions? For example, a long chain with many bodies in it. Do the authors have an intuition or maybe have tested this method in bigger settings?
* Have the authors tried to use an MLP instead of a GRU for the node embeddings, as in the original NRI paper? Is this an important architectural choice?
* What if the pseudo-labels are poorly estimated? Then the joint network would have bad labels and might result in a decrease in accuracy. How difficult is to train this pipeline?
* Line 131: "...from raw trajectories, These features..." might refer to "...from raw trajectories. These features...".
* Equation 5: Closed parenthesis is missing.

**Ethical Concerns:**

["NO or VERY MINOR ethics concerns only"]

**Final Justification:**

My main concerns were the limited applicability to toy datasets and arbitrary architectural choice, and those were satisfactorily addressed in the rebuttal with two additional experiments (DoSI benchmark dataset and MLP ablation study). The authors also provided the computational time overhead with respect to the baseline method, and solved all the minor issues pointed out. Based on the author's rebuttal, I've decided to rise the initial score to an accept.

**Limitations:**

Limitations are addressed appropriately.

**Paper Formatting Concerns:**

There are no formatting concerns.

**Quality:**

3

**Strengths And Weaknesses:**

Strengths
* The technique is valid with both labelled and unlabelled data in the latent space.
* The method is tested over 4 different baselines and ablation studies.
* The code is provided (not publicly available yet).

Weaknesses
* The results are modest with respect to the base architecture (MPM).
* Usually in real world applications the latent space labels are unknown, and in that case (supervision via pseudo-labels) the method is quite pointless even though the results are marginally better than the vanilla architecture.
* There is no mention about the computational overhead of the iterative process and the addition of an extra network.
* The study is restricted to small systems with constant edge interactions ($\sigma$) and fixed graph topology.

---

> ### Author Rebuttal · Authors · 2025-07-31
>
> We sincerely thank the reviewer for the thoughtful and constructive feedback. Below, we address each point raised and provide clarifications and additional evidence where needed.
>
> > **W1**: The results are modest with respect to the base architecture (MPM).
>
> MPM is a strong baseline on the synthetic physical simulation dataset. While the improvements achieved by the MPM-based PreBoot-SI appear small in absolute terms (e.g., from 93.3\% to 94.3\% on the charged dataset), we would like to point out that such improvements are meaningful in context: this corresponds to a 15% relative reduction in error rate (from 6.7% to 5.7%). Moreover, for structural inference tasks in complex situations, perfect recovery is often unattainable, so even marginal accuracy gains could be meaningful.
>
> In addition, we conducted supplementary experiments on the benchmark DoSI [1], and on three dataset with real-world structure in DoSI: Brain Networks (BN), Gene Regulatory Networks (GRN) and Vascular Networks (VN). The performance improvement of PreBoot-SI compared to the MPM baseline was more significant. The following table summarizes the improvement on DoSI benchmark, each dataset's interaction graph contains 15 nodes, and the performance metric is the AUROC value:
>
> |                 | Springs BN | Springs GRN | Springs VN | NetSims BN | NetSims GRN | NetSims VN |
> |-------------------------|------------|-------------|------------|------------|-------------|------------|
> | MPM                     | **99.98**  | 94.02       | 96.56      | 99.95      | 76.06       | 91.18      |
> | PreBoot-SI (MPM-based)  | **99.98**  | **98.95**   | **99.39**  | **99.97**  | **95.32**   | **96.75**  |
>
> DoSI also includes results of other VAE-based structural inference models. PreBoot-SI based on MPM has achieved state-of-the-art results on these three datasets. Due to space limitations, we kindly refer the reviewer to our response to Reviewer YANt, 6Y2C or vjqL for detailed results and discussion.
>
> [1] Benchmarking Structural Inference Methods for Interacting Dynamical Systems with Synthetic Data. In NeurIPS Datasets and Benchmarks Track, 2024.
>
> > **W2**: Usually in real world applications the latent space labels are unknown
>
> Similar to NRI [1], we also target situations where the real structure is unknown. To address this, we propose two complementary methods to obtain pseudo labels instead of directly using ground-truth structure labels:
>
> (a) With prior knowledge of dynamics: We simulate a synthetic meta-dataset comprising systems with similar dynamics (e.g., radial force interactions) but varying parameters. These synthetic systems are generated with structural labels, and the training on them is intended to capture common structure–trajectory patterns shared across systems with similar dynamics.
>
> (b) Without prior knowledge of dynamics: We use a baseline model (e.g., NRI) to generate initial pseudo-labels, which may be inaccurate but still serve as informative priors. Training the baseline model also does not require ground-truth structure labels.
>
> In both cases, PreBoot-SI progressively refines the structural estimates over iterations.
>
> [1] Neural Relational Inference for Interacting Systems, In ICML, 2018.
>
> > **W3**: There is no mention about the computational overhead of the iterative process and the addition of an extra network.
>
> We acknowledge that iterative training introduces extra computation, which mainly derives from iterative training of $\text{SI}_{\text{joint}}$. The following table shows the training time on Charged Particle dataset on a single 3090 Ti GPU (the complete computational cost will be added in the supplementary materials). PreBoot-SI requires 2–3 times more training time compared to the baseline model and remains within practical limits. Additionally, as mentioned in Section 6, we plan to explore sparsity-guided encoders that potentially reduce complexity from $\mathcal{O}(N^2)$ to $\mathcal{O}(N)$.
>
> |   | $\text{SI}_{\text{prior}}$|  $\text{SI}_{\text{joint}}$ | PreBoot-SI | NRI baseline |
> |------------|---------------------|---------------------|-------------|---------------|
> | 5 Objects  | 5 min               | 1h                  | 6h          | 2.5h          |
> | 10 Objects | 15 min              | 5h                  | 22h         | 10h           |
>
> > **W4 & Q1**: The study is restricted to small systems with constant edge interactions $\sigma$ and fixed graph topology. How scalable is this approach to bigger systems and sparser interactions? For example, a long chain with many bodies in it. Do the authors have an intuition or maybe have tested this method in bigger settings?
>
> We agree that evaluating only 5- and 10-node systems with fixed topologies and interaction strength $\sigma$ is a limitation. In this work, our primary goal is to introduce a general-purpose performance enhancement framework for structural inference, rather than to target specific scalability improvements. To ensure fair and direct comparison with prior methods, we adopted standard benchmark settings from previous work ([1], [2]).
>
> Meanwhile, the primary scalability bottleneck lies in the fully connected GNN encoder employed by NRI and MPM, not in the PreBoot-SI framework itself. As noted in Section 6, our method is compatible with sparse or dynamic encoders, and PreBoot-SI has the potential to utilize learnable structural priors to initialize sparse GNN encoders. In future work, we aim to integrate PreBoot-SI into constructing sparse GNN graphs, enabling inference on longer chains or large-scale systems.
>
> [1] Neural Relational Inference for Interacting Systems, In ICML, 2018.
>
> [2] Neural Relational Inference with Efficient Message Passing Mechanisms, In AAAI, 2021.
>
> > **Q2**: Have the authors tried to use an MLP instead of a GRU for the node embeddings, as in the original NRI paper? Is this an important architectural choice?
>
> We chose to use GRU instead of MLP for node embeddings, particularly in the $\text{SI}_{\text{prior}}$ module trained on the synthetic meta-dataset. This decision is motivated by the nature of the pretraining task: the meta-dataset is designed to capture recurring trajectory-structure patterns across systems with similar dynamics. For example, if two objects consistently move away from each other across time, this may suggest a repulsive edge, which is a temporal pattern that GRUs are well-suited to capture.
>
> To isolate the architectural effect, we also replaced the MLP-based embedding in the original NRI baseline with a GRU. This change resulted in only negligible performance differences, suggesting that the gains from PreBoot-SI arise primarily from the overall mechanism rather than the choice of node embedding. A brief comparison is provided below, it demonstrates the performance changes of NRI-based PreBoot-SI when replacing node embeddings on Charged dataset (5 nodes), as analyzed above, the replacement of GRU embedding mainly affects the situation with dynamic prior where synthetic meta-dataset is used.
>
> | Model Type                          | Edge prediction accuracy (%) |
> |------------------------------------|-------------------------------|
> | NRI (GRU)                           | 81.9                          |
> | NRI (MLP)                           | 82.2                          |
> | PreBoot-SI (w/o prior, GRU)         | 89.3                          |
> | PreBoot-SI (w/o prior, MLP)         | 88.9                          |
> | PreBoot-SI (w/ prior, GRU)        | **91.2**                         |
> | PreBoot-SI (w/ prior, MLP)        | 89.1                          |
>
> > **Q3**: What if the pseudo-labels are poorly estimated? Then the joint network would have bad labels and might result in a decrease in accuracy. How difficult is to train this pipeline?
>
> As shown in Figure 3(b), PreBoot-SI remains robust even when initialized with extremely poor pseudo-labels. For example, when we provide randomly generated labels with 50\% accuracy (i.e., equivalent to random guessing for two edge types), the model still converges to the performance ceiling of NRI baseline.
>
> This suggests that the bootstrapping process can effectively correct noisy supervision, and that training remains stable and straightforward even when pseudo-labels are poorly estimated.
>
> > **Q4, 5**: Line 131: "...from raw trajectories, These features..." might refer to "...from raw trajectories. These features...".
> Equation 5: Closed parenthesis is missing.
>
> Thank you for pointing them out. We will correct:
>
> Line 131: "...from raw trajectories, These features..." $\rightarrow$ "...from raw trajectories. These features...".
>
> Equation (5): we will add the missing closing parenthesis.
>
> Once again, we thank the reviewer for the valuable comments. We believe these clarifications and additional results strengthen the case for our paper and demonstrate the practical potential of PreBoot-SI.

---

> > ### Comment · Reviewer_FSLU · 2025-08-03
> >
> > I thank the authors for the rebuttal and clarifying many of my concerns. I have several comments:
> > * W1: I appreciate the addition of the benchmark examples, I think it strengthens the claims of the paper.
> > * Q2: The MLP experiments are interesting and show that the results are mostly agnostic to the network used, unless strong temporal correlations are present.
> >
> > Based on the comments above, the addition of the new experiments and the address of the previous comments, I’ve decided to rise the initial score to an accept.

---

> > > ### Author Response · Authors · 2025-08-04
> > >
> > > We sincerely thank the reviewer for the thoughtful and constructive feedback. We greatly appreciate your recognition of our work and your valuable comments, which helped us further improve the clarity and rigor of the paper. We are especially encouraged by your decision to raise the score and by your positive remarks on the added benchmark examples and the MLP experiments. Based on your suggestions and the discussion during the rebuttal, we will revise and enrich the manuscript accordingly. Furthermore, we plan to explore the generalization of PreBoot-SI on large-scale and real-world datasets in our future work.

---

### Decision · Program_Chairs · 2025-09-17

**Decision:**

Accept (poster)

**Comment:**

Five knowledgeable reviewers recommend Accept, and I recommend Accept. The authors provided new experimental evaluations for their method during the rebuttal and clarified several claims, which have improved the quality of the work. I ask the authors to include these additional experiments and discussion in the Camera Ready version.